

# Samples and data accessibility in research biobanks: an explorative survey

Marco Capocasa[1,*], Paolo Anagnostou[1,2,*], Flavio D'Abramo[3,*], Giulia Matteucci[1], Valentina Dominici[1,2], Giovanni Destro Bisol[1,2] and Fabrizio Rufo[1,2]

[1] Istituto Italiano di Antropologia, Rome, Italy
[2] Department of Environmental Biology, Sapienza University of Rome, Rome, Italy
[3] Charité Comprehensive Cancer Center, Berlin, Germany
[*] These authors contributed equally to this work.

## ABSTRACT

Biobanks, which contain human biological samples and/or data, provide a crucial contribution to the progress of biomedical research. However, the effective and efficient use of biobank resources depends on their accessibility. In fact, making bio-resources promptly accessible to everybody may increase the benefits for society. Furthermore, optimizing their use and ensuring their quality will promote scientific creativity and, in general, contribute to the progress of bio-medical research. Although this has become a rather common belief, several laboratories are still secretive and continue to withhold samples and data. In this study, we conducted a questionnaire-based survey in order to investigate sample and data accessibility in research biobanks operating all over the world. The survey involved a total of 46 biobanks. Most of them gave permission to access their samples (95.7%) and data (85.4%), but free and unconditioned accessibility seemed not to be common practice. The analysis of the guidelines regarding the accessibility to resources of the biobanks that responded to the survey highlights three issues: (i) the request for applicants to explain what they would like to do with the resources requested; (ii) the role of funding, public or private, in the establishment of fruitful collaborations between biobanks and research labs; (iii) the request of co-authorship in order to give access to their data. These results suggest that economic and academic aspects are involved in determining the extent of sample and data sharing stored in biobanks. As a second step of this study, we investigated the reasons behind the high diversity of requirements to access biobank resources. The analysis of informative answers suggested that the different modalities of resource accessibility seem to be largely influenced by both social context and legislation of the countries where the biobanks operate.

Corresponding author
Marco Capocasa,
marco.capocasa@uniroma1.it

## INTRODUCTION

Biobanks play a crucial role in the biological research involving human subjects and provide a fundamental contribution to the rapid growth of scientific endeavour. This has been well demonstrated, particularly in the past ten years, by the investments made by many countries in order to build such infrastructures and to manage the biological resources they store (*Kaye, 2011*). Biobanks hold human biological samples and/or data

to facilitate research over time (*Wolf et al., 2012*). Their development across the world, together with improvements in laboratory technologies, have dramatically increased opportunities to study collections of bio-specimens (and their related data) with broader perspectives than those possible by small collections maintained by single research groups (*Haga & Beskow, 2008*). However, in addition to the creation of these new opportunities, the rapid evolution taking place in the biobanking field has created new challenges for researchers due to the huge potential benefits of having access to biological resources.

In order to draw a more detailed picture of how biobanks manage their resources, as well as considering the relationships (and even the contradictions) between the material and the informational spheres of biological samples, we must take into account the propensity of these institutions to share bio-specimens and data across scientific communities. The first challenge for biobanks consists in finding an equilibrium between the scientific interests of researchers and the expectations of donors. This can be reached by better exploiting the capabilities and flexibility of current forms of informed consent (*Kaye, 2012*; *Macilotti, 2013*; *Colledge et al., 2014*; *D'Abramo, 2015*). However, the design of an informed consent able to guarantee the sustainability of resource availability does not solve the issues related to the economic interests usually hidden behind the scientific research. This is the case of several web services that sell direct-to-consumer genetic tests. Through their activities, these companies accumulate large amounts of samples and data that however, remain unavailable to most research communities and groups (e.g., deCODEme, 23andme, Navigenics; see *Knoppers, 2010*). Finally, even if biobanks embrace the open science principles, many bioethical issues can emerge as sample and data sharing policies are different from country to country. In fact, the existence of local legislation ensures compliance with habits and values characterizing the socio-cultural contexts in which biobanks operate (*Kaye, 2006*; *Haga & Beskow, 2008*). On the other hand, a widespread and efficient sharing of bio-resources from different countries can only be assured through the achievement of a global consensus on the legislation, the standards and the modalities to be followed. Starting from the preparation of informed consent, the biobank staff must take into account a number of issues when planning the management of the samples and data. They have to meet the requirements imposed by ethics committees, overcome the difficulties in explaining the future uses of existing samples and put the potential donor in a condition that will allow him to make a really informed decision (*Colledge et al., 2014*; *D'Abramo, Schildmann & Vollmann, 2015*).

Given these premises, it cannot be denied that the progress of human biological research largely depends on the openness of resources and scientific knowledge. In fact, making bio-resources promptly accessible to everyone could favour the common good. Furthermore, optimizing their use, controlling quality and promoting general scientific creativity will guarantee a more rapid and efficient progress of bio-medical research (*Fischer & Zigmond, 2010*; *Trinidad et al., 2010*; *Milia et al., 2012*; *Oliver et al., 2012*). Although this has become a rather common belief, several laboratories are still secretive and continue to withhold samples and data (*Nelson, 2009*; *Cadigan et al., 2014*). The scientific and academic interests of researchers are important, but they also have responsibilities towards the tax paying public. In fact, the scientific community often regards biobanks and their

services as simple source of material for the research and forget that the sample come from human subjects. *Milanovic, Pontille & Cambon-Thomsen (2007)* have clarified this concept, defining biobanks as "ambiguous entities" that "might be presented as places for archival storage of a cultural patrimony freely accessible for relevant activities, or as commercial enterprises with lucrative potential." At the same time, biobank donors have also raised concerns about the fact that, in particular conditions, private and commercial interests in biobanking may prevail over the public good and this could lead to social tensions (*Godard et al., 2010*). The importance of identifying solutions which satisfy the needs of both researchers and citizens is well testified by the engagement of a political economic structure such as the *Organisation for Economic Co-operation and Development* (OECD) in supporting open access to public funded research products (*Arzberger et al., 2004*).

Previous studies conducted on European and U.S. biobanks have provided information on the developing trends of biobanking, giving detailed pictures of the type of sample and data stored therein (*Hirtzlin et al., 2003*; *Zika et al., 2011*; *Henderson et al., 2013*). Other studies have investigated the opinion of participants and the public about the relationships between sample and data sharing practices and privacy concerns (*Kaufman et al., 2009*; *Lemke et al., 2010*). However, to date, only a limited number of studies have faced the issue of sample and data sharing behaviour of research biobanks (e.g., see *Milanovic, Pontille & Cambon-Thomsen, 2007*; *Pereira, 2013*). The present work aims to investigate sample and data accessibility in research biobanks operating all over the world by means of a questionnaire-based survey. We observed a low rate of free accessibility for both data and biological samples while the requirements for accessing to the non-open resources were found to be highly heterogeneous. In order to evaluate the reasons behind this heterogeneity, we analysed the relationships between sharing strategies and legal frameworks of the countries in which biobanks operate.

## MATERIALS & METHODS

In this study, we considered the definition of "biobank" as a repository that stores human biological samples, with or without linking them to genetic or clinical data (see *Haga & Beskow, 2008*). Therefore, we have not taken into account non-human bio-repositories or on-line databases. The online survey was administered to a total of 238 biobanks (see Table 1) operating in Europe (95), America (104), Asia (25), Africa (2) and Oceania (12). The biobanks were selected in February 2014, through the use of three keywords ("biobank," "research biobank" and "human biobank") and three search engines (Google, Google Scholar and Pubmed). Firstly, we used the generic keyword "biobank" obtaining around 550,000 results with Google, 25,000 with Google Scholar and 2,100 with Pubmed. To refine our search, we added the term "research" and obtained around 16,700 results with Google, 428 with Google Scholar and 25 with Pubmed. Furthermore, we performed a second refinement adding the term "human" and obtained 4,500 results with Google, 284 with Google Scholar and 19 with Pubmed. We then inspected all these latter results and identified the ones that refer to research biobank sites from which we recorded their contact emails. This keyword-based procedure was adopted in order to select a random sample of biobanks that could be easily found by anyone (researchers and the public).

Table 1 **Geographic distribution of biobanks involved in this study.** Percentage of respondents in brackets.

| Continent | Country | Biobanks | |
| | | Invited | Responded |
| --- | --- | --- | --- |
| Africa | South Africa | 1 | 0 (0) |
| | Zimbabwe | 1 | 0 (0) |
| America | Brazil | 1 | 0 (0) |
| | Canada | 12 | 2 (16.7) |
| | USA | 91 | 14 (15.4) |
| Asia | China | 2 | 0 (0) |
| | India | 3 | 0 (0) |
| | Iran | 1 | 0 (0) |
| | Israel | 3 | 1 (33.3) |
| | Japan | 4 | 1 (25.0) |
| | Korea | 1 | 0 (0) |
| | Malaysia | 3 | 0 (0) |
| | Qatar | 1 | 0 (0) |
| | Singapore | 4 | 0 (0) |
| | Taiwan | 1 | 0 (0) |
| | Thailand | 2 | 0 (0) |
| Europe | Austria | 4 | 2 (50.0) |
| | Belgium | 3 | 1 (33.3) |
| | Estonia | 1 | 1 (100) |
| | Finland | 1 | 1 (100) |
| | France | 7 | 2 (28.6) |
| | Germany | 19 | 3 (15.8) |
| | Greece | 2 | 0 (0) |
| | Hungary | 1 | 0 (0) |
| | Iceland | 1 | 0 (0) |
| | Ireland | 3 | 1 (33.3) |
| | Italy | 12 | 5 (41.7) |
| | Latvia | 1 | 0 (0) |
| | Luxembourg | 1 | 0 (0) |
| | Malta | 1 | 0 (0) |
| | Netherlands | 4 | 1 (25.0) |
| | Norway | 2 | 1 (50.0) |
| | Poland | 1 | 0 (0) |
| | Portugal | 1 | 0 (0) |
| | Spain | 5 | 1 (20.0) |
| | Sweden | 5 | 1 (20.0) |
| | Switzerland | 4 | 0 (0) |
| | United Kingdom | 16 | 6 (37.5) |
| Oceania | Australia | 12 | 2 (16.7) |
| | Total | 238 | 46 (19.3) |

The questionnaire was compiled in order to obtain a detailed picture of the sampling activities, the sample and data accessibility criteria and the legal frameworks for their access. The final part of the questionnaire was based on a preliminary analysis of twenty biobanks selected following a geographic criterion (9 European, 3 North-American, 2 South-American, 3 Asian and Australian biobanks). The preliminary analysis was conducted by contacting each biobank asking for explanations regarding their sample and data sharing modalities. We asked them to provide information replying to our e-mail and/or by unstructured telephone interviews. Five biobanks responded to our request and with two of them, we also conducted the interview. Furthermore, we analysed their web sites in order to verify the presence of specific information about these aspects. Finally, we used the collected information to build the questionnaire of the present study. The questionnaire consisted of 21 questions (9 closed and 12 open-ended) organized into three sections (see File S1). The first section (General information) refers to the name and the place where the biobank operates and other information regarding funds, the sampling criteria adopted and the type of biological resources stored (sample and/or data). The second section (Biological samples) investigates the sample collection, the ethical requirements and the legal framework to which the biobank refers to for the management of accessibility to biological samples. The last section (Data) includes questions regarding the data collection and the legal framework to which the biobank refers to for the regulation of data accessibility. The questionnaire was built and administered using Google Forms (http://www.google.com/forms/about/) and survey participation was requested by contacting each respective biobank address by e-mail, explaining the aims of our research. We launched our survey on 18th April 2014 and sent three reminders (28th April, 5th May and 19th May, 2014), closing it at the end of May 2014. As previously stated in the invitation form, the administration of the questionnaire was carried out anonymously. Neither personal information nor the names of biobank were disclosed to others in managing the dataset.

The descriptive statistics of the answers to the closed questions were obtained using Microsoft Excel 2010. Open questions were analysed considering the clarity and informative nature of the answers subdividing them into three categories: exhaustive answer (it provides a clear and complete explanation of the question); partial answer (some information is missing); elusive or no answer (it does not provide any of the information requested). Furthermore, since many of the answers provided links to external resources (e.g., web links), we also evaluated the exhaustiveness of these documents in order to acquire the information needed to fulfil the questions. When the external references failed to provide clear information, depending on retrievability difficulties or language issues (the replies were written neither in English nor in Italian), we classified the answer as partial or elusive. Data was uploaded as online supporting information (File S2) and deposited in Zenodo (DOI 10.5281/zenodo.17098).

## RESULTS

### General information of the responded biobanks

Overall, we obtained responses from 46 out of 238 biobanks (19.3%): 26 in Europe, 16 in America, 2 in Asia and 2 in Oceania (Table 1). Most of the participant institutions are
from United States (30.4%, $n = 12$) followed by the United Kingdom (13.0%, $n = 6$), Italy (10.9%, $n = 5$) and Germany (6.5%, $n = 3$).

More than half of the 46 biobanks are publicly funded (58.7%), whereas 23.9% and 17.4% make use of private (both profit-making and non-profit-making) funds or both types of funds, respectively. Interestingly, some continental variations may be observed. In Europe, the rate of institutions that receive only public funds is three times higher than that observed on the American continent (73.1% and 25.0%, respectively) whereas the percentage of biobanks that make use of both types of funding is not significantly different (19.2% and 18.7%, respectively). All the Asian and Australian biobanks analysed here are only publicly funded. However, the low number of institutions that responded to our survey (only 4) makes any form of comparison with the other continents difficult.

Looking at the sampling criteria used by biobanks to collect bio-specimens, most of them focused their attention only on disease-based samples (41.3%) or coupling it with other criteria such as type of tissue (17.4%) or the geographic area where the samples were collected (8.7%). Only seven biobanks focus their attention on criteria other than diseases. Three consider geography (6.5%), two types of tissue (4.3%), and two institutions concentrate on a population-based-sample collection (4.3%).

Regarding storing activities, a wide range of biological materials have been collected by the sampled biobanks (e.g., blood, plasma, serum, urine, saliva, nucleic acids, cell lines). Eight institutions store only bio-specimens and operate in the United States (3), in Europe (3; Italy, Sweden and United Kingdom) in Asia (2). The remaining 38 biobanks store both biological samples and data (89.1%).

## Bio-specimens collection and accessibility: legal and ethical aspects

In the first open question, we asked for the ethical requirements followed by the biobank for the sample collection procedures (Question B2; see Fig. 1). We mostly focused on the consent obtained from participant (if any) and on approval by a third party (e.g., Ethics Committee, Institutional Review Board (IRB)). Twenty-two biobanks (47.8%) provided informative answers, referring, in all the cases, to the consent procedures and often pointing to guidelines, specific local or international laws, and approval by ethics committees or institutional review boards. Open consent (through which participants give authorization to researchers for a broad range of projects) seems to be the most utilized approach to involve donors. On the other hand, informed, specific consent (in which participants authorize single projects whose aims, benefits and risks should be well described, and through which biobanks should ask participants to give permission again for every new project involving their samples and/or data) was found to be be frequent. Some biobanks provide information on privacy issues describing, in most cases, the anonymized characteristics of the personal data and the fact that they comply with national and federal laws on data protection. Very few answers stressed the possibility for donors to pull out of the biobank research (opt-out option). We categorized 15 answers (32.6%) as semi-informative since they only refer to third party responsibility for the sample collection procedures, without mentioning any other description regarding the type of consent used (waived or presumed consent included), or else when a reference to specific documents was made (e.g., certifications or

laws) but this was not easily readable/accessible. Nine answers (19.6%) were not informative because they were either left blank or referred to vague documents/criteria.

Concerning the strategies of collection and storage of biological samples, we found 24 biobanks (52.2%) that do not accept samples from external research groups, with roughly similar percentages in European (41.7%) and American (50.0%) continents. On the contrary, 22 biobanks (47.8%) also store biological samples collected by external research groups. Sixteen of them operate in Europe (72.7%), 4 in America (18.2%), 1 in Asia (4.5%) and 1 in Oceania (4.5%). The majority of them (86.4%), in order to accept samples for storage from external groups, ask the latter to respect the same ethical requirements adopted by the biobank itself in its sampling procedures. All the biobanks analysed make it possible for researchers to gain access to their bio-specimen collection. Among them, only 2 biobanks (1 European (Estonia) and 1 American (USA)) offer free and unconditioned accessibility to their samples, whereas the remaining 95.7% (44 out of 46) require specific conditions to be satisfied in order to give permission to access their samples. However, our request for specifications regarding the accessibility criteria (Question B4.1; see Fig. 1) obtained only 12 informative answers in which at least one criterion has been indicated. The analysis of these answers highlights how sample accessibility seems to be linked to whether the applicants specify their research aims (e.g., studies on a defined disease) and/or the origin of research funds (public, private or both). We also considered those answers indicating that samples are for sale to be informative, or when one of the above-listed criteria were specified and readable in external links provided in the answer. Among the answers, 25 were classified as semi-informative. We defined the answers as semi-informative when it was indicated that access is decided by third parties (e.g., IRBs, ethics committees), when a vague criterion was stated (e.g., research project relevance, or researchers working in the public interest), when specific agreements were indicated but the description was difficult or impossible to read (i.e., in languages other than English or Italian), or when the biobank institution had a person responsible for the access to biological samples. Finally, we categorized 7 answers as not informative because they were left blank or because they refer to agreements or documents that are totally inaccessible.

More than half of the biobanks (54.3%) refers to a specific legal framework for access to their biological samples. However, only 16 biobanks (34.8%) provided informative answers showing that there are no shared standards but different approaches influenced by the social context in which they operate (Question B5.1; see Fig. 1). The possibility to gain access to samples seems to depend mainly on the approval of ethical committees, scientific bodies or bilateral agreements (some biobanks also provided information regarding the model followed, e.g., OECD recommendations or legal contracts that concern customs laws regarding the commercial circulation of biological materials). However, national laws (e.g., the Italian *Garante della Privacy*, German Data Protection Laws, Spanish Act 14/2007 and Spanish Royal Decree 1716/2011), common international regulations (i.e., the European legal framework), or criteria indicated in international agreements should also be taken into consideration when a request for access to a collection of biological samples is presented. Concerning the possibility of finding documents relative to the

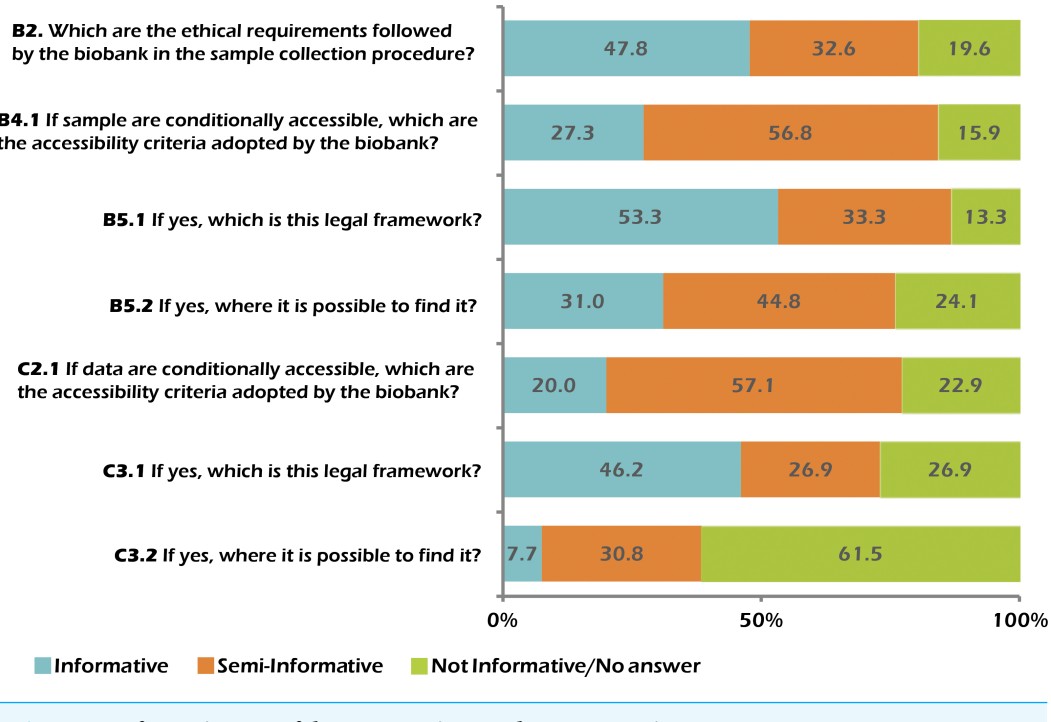

**Figure 1** Informativeness of the answers given to the open questions.

legal framework followed by the biobanks, only 31% provided detailed information (Question B5.2; see Fig. 1).

## Data collection and accessibility: legal and ethical aspects

Most of the surveyed biobanks (41 out of 46; 89.1%) store data extracted from the analysis of their own samples. Differently from what we observed for the biological samples, 23 out of 41 biobanks (56.1%) also store data produced by external research groups that have used their samples. Most of them (73.9%) offers this service only if the external research groups follow the same same legal framework of the biobank in question.

A slight difference between sample and data sharing propensity is evident when considering their degree of accessibility. In fact, 3 biobanks (7.3%; 2 Americans (USA) and 1 European (Sweden)) do not allow any access to their data, whereas 3 other biobanks (7.3%; 1 American (USA) and 2 Europeans (France and Italy)) claim to follow a strict open data policy, giving completely free access to their data. However, similarly for bio-specimens, the majority of biobanks (35 out of 41; 85.4%) allow external research groups to access their data in compliance with certain conditions. We asked them which accessibility criteria they adopt (Question C2.1; see Fig. 1). Seven of these biobanks (20%) gave us informative answers, describing codification procedures, providing reference to specific guidelines, giving access to projects focused on specific groups of diseases, or stating clear criteria (e.g., co-authorship). Twenty of the biobanks (57.1%) provide semi-informative answers which refer to (i) third party authorization for data access (i.e., ethics committees, IRBs, scientific boards); (ii) criteria linked to the decision of a single researcher within

the biobank institution; (iii) external documents that do not clearly state the criteria adopted for data sharing; (iv) other vague criteria (e.g., application for data access through a letter of intent). Finally, among the biobanks giving conditional access, 8 (22.9%) were uninformative because their answers were inappropriate or because the criteria were particularly vague (e.g., access given to authorized personnel, access given for research made in public interest).

Similarly to what we observed for bio-specimens, also for data accessibility, the majority of the biobanks (57.1%) refer to a variety of legal frameworks, depending on the legislation of the country where they mostly operate. Among the 35 biobanks (76.1%) that grant conditional data accessibility, 12 provided informative answers regarding this topic (Question C3.1; see Fig. 1). They generally referred to national and international legal frameworks and agreements (e.g., European legal framework, Material Transfer Agreement and the Health Information Privacy and Portability Act in the USA). Interestingly, only one biobank highlighted the role of the privacy guarantor for personal data protection in this procedure and only two biobanks (7.7%) provided the web link necessary to access to the legal framework documents (Question C3.2; see Fig. 1).

## DISCUSSION

In this study, we explore how and at what level data and biological samples stored in research biobanks are accessible and reusable. Primarily, most of the biobanks who responded to our survey give access to their samples and data. However, free and unconditioned accessibility is not common practice. In fact, external research groups eager to use biobank resources must satisfy specific conditions in order to receive samples and gain access to databases. Unfortunately, most of the biobanks we contacted provided vague or difficult-to-read information about their accessibility criteria. This is an important result which shows that there is still little clarity and a certain reluctance to share scientific resources. This lack of sharing contrasts sharply with the emerging dependence of biomedical research on the activities of biobanks (*Kaye, 2011*; *Kaye et al., 2015*). Nonetheless, this reluctance in sharing contradicts the latest European research programme, Horizon 2020, where specific policies for open data and open access are foreseen (*Leonelli, Spichtinger & Prainsack, 2015*). Our analysis of the informative answers points to three major issues related to the accessibility of biobank resources.

Firstly, applicants are requested to explain what they would like to do with the required resources. This information is closely related to the specific data/bio-specimen sharing clause contained in the original consent form. At the same time, it provides a certain degree of control by the biobanks over the credentials and scientific reputation of the user and his research group. Verifying the reliability and seriousness of applicants and minimizing the misuse of data and samples is fundamental for biobanks. The ethical and technical approach of scientific-resource management can promote public trust in the work of these institutions, thus increasing willingness to participate in their activities (*De Robbio, 2010*).

Secondly, availability, amount and origin (public or private) of research funds are aspects involved in the establishment of fruitful collaborations between biobanks and research labs. Publicly funded research seems to be preferred over studies granted by

private (both profit and non-profit-making) funding bodies. This result complies with the recommendation towards Open Access of scientific resources produced with public funds proposed by the Organisation for Economic Co-operation and Development (OECD) in its report "ECD principles and guidelines for access to research data from public funding" published in 2007 (see *Pilat & Fukasaku, 2007*). The OECD report highlights the social, non epistemic, value of "public good" in sharing and the importance of public scientific research and investment. However, only one of the biobanks surveyed explicitly used the concept of sharing as "common good." On the other hand, scientific resource sharing also has epistemic values regarding scientific rigor in favor of scientific progress and this approach can guarantee a more "effective and transparent biobank practice" (*Demir & Murtagh, 2013*). Not only the source but also the availability of funds needed to carry out the criteria research and payment for access to samples and data were found to be fundamental criteria adopted by biobanks when deciding whether or not to make their data available to third parties. In fact, the presence of clauses directly related to economic benefits for biobanks reveals their possible "second nature" as profit-making institutions that offer services concerning the collection and storage of biological samples and access to this material for researchers. Thus, we can assume a relationship between private funds, buying and selling of biobank resources and the widespread sharing of data and biological samples. However, it is unclear if the commercial nature of biobanks is really a barrier to sharing. *Caulfield et al. (2014)* suggest that the sharing of data and samples is a practice that "may be impacted or hindered by the introduction of private funding and collaboration with private entities, as the expectations of private entities and agreements governing such partnerships may create sharing barriers." Other authors sustain that venture capitalism, with its continuously fluctuating expectations, have a strong impact on open data, above all when the boundaries between successful data sharing and unfruitful initiatives become difficult to recognize, making financial investments, in these kinds of scientific enterprise, risky but potentially rewarding (*Leonelli, 2013*).

Thirdly, we found that recognition of co-authorship is a requirement for some biobanks in order to grant access to their data. *Tenopir et al. (2011)*, in their study on the data sharing practices and perceptions of scientists, also reported the request for co-authorship on a publication as a fair condition for the use of data produced by others. A similar result was also found by *Milanovic, Pontille & Cambon-Thomsen (2007)* in their empirical study on the sharing of biological samples and data in biosciences. This kind of request falls within the broader context of the management of scientific resources in order to gain advantages in academic competition. According to *Vogeli et al. (2006)*, this behavior may contribute to spreading a climate of mistrust and lack of cooperation within the scientific community.

To sum up, these results suggest that economic and academic aspects are involved in determining the extent of sharing of samples and data stored in biobanks. There is a consolidated habit whereby biobank professionals mostly concentrate on commercial aspects whereas researchers are more interested in academic pursuits (*Pereira, 2013*). Fortunately, these detrimental attitudes for scientific progress and for the ethics of science cannot be generalized. In fact, the culture of open science has begun to spread over the past decade in different fields of life sciences (see *Destro Bisol et al., 2014a*;

*Destro Bisol et al., 2014b* and related citations therein). More specifically, scholars and researchers increasingly perceive the sharing of scientific resources as a primary requirement for the development of new opportunities for collaboration (e.g., see *Foster & Sharp, 2007*; *Fischer & Zigmond, 2010*; *Boulton et al., 2012*; *Mauthner & Parry, 2013*). In the case of research involving human subjects, data and sample sharing practices have been carried out following different protocols. All these protocols face obstacles and restrictions due to both practical (e.g., the definition of informed consent) and ethical issues (e.g., privacy and confidentiality concerns, prediction of potential reuses; see *Blumenthal et al., 2006*; *Teeters et al., 2008*; *Institute of Medicine (IOM), 2015*). Moreover, given the different nature of data and samples, they do not necessarily follow identical sharing procedures. In fact, while data sharing culture in biosciences seems to be catching on among both researchers and policymakers, the same cannot be said for samples (*Pereira, 2013*). However, *Pereira (2013)* depicts a more optimistic view about the willingness to share biological resources by biobank professionals, highlighting that they "showed considerable interest in advancing research and a generally altruistic perspective toward sharing samples and making materials accessible to the research community." One good practice could consist in disclosing the origin of funding and the aim of the research considering that today we are in an era in which the characteristics of public research are ever more similar to those of private commercial science (see *Jasanoff, 2002*; *Krimsky, 2003*; *Krimsky, 2005*)—e.g., openness and transparency of claims and evidence substituted by secrecy (*Ledford, 2014*), fabrication of results and bias against negative results (*Fanelli, 2012*). It is useful to remember that the 'bank' metaphor overcomes the notion of "bio-repositories" or "bio-libraries" (*Schneider, 2008*) and that biobanks can diverge diametrically in objectives and outcomes, or diametrically divergent visions and practices can coexist within the same biobank. In this respect, biobanks are often regulated by national and communitarian trade laws that hinder harmonization (i.e., bilateral agreements such as the Transatlantic Trade and Investment Partnership between Europe and the USA) and this could influence or divert local interests and national health services, as well as medical research and local biotech companies.

In the second step of our study, we investigated the reasons behind the observed high heterogeneity of the requirements to gain access to the biobanks' resources. Most of the surveyed biobanks adopted specific legal frameworks that researchers should take into consideration in order to gain access to samples and data. The comparison of the information obtained from the biobanks highlighted that these institutions follow different standards and procedures regarding the sharing of their biological samples and data. The different modalities of resource accessibility seem to be highly influenced by social context and legislations of the countries where the biobanks operate. The fact that only few biobanks provided informative answers about this topic could be interpretable as strong evidence that resource sharing is still a cumbersome practice. This lack of clarity raises both ethical and practical issues: how to implement the sharing of ethical conditions linked to the use of data and biological samples. A first practical step could be the opportunity for donors to make their own choices through the informed consent process. The ethical principles at the basis of informed consent in research involving human subjects (i.e., respect, individual autonomy, protection of privacy) are inalienable and their importance is even

more evident in the case of biobanks due to their nature of institutions involving multiple researchers within multiple research projects (*Fullerton & Lee, 2011*). But, precisely due to their nature, "it is difficult to obtain consent for all future research uses at the time of recruitment into the biobank or before such research commences" (*Kaye et al., 2015*) a requirement specifically stated in the seventh revision of the Declaration of Helsinki of 2013 (*World Medical Association, 2013*). As declared by Jane *Kaye et al. (2015)*, classical informed consent is an inefficient tool in the attempt to overcome the obstacles in data and samples sharing due to its static, paper-based format, which is generally only recognized at national level. Furthermore, we must bear in mind that, particularly in the European context, privacy laws make the possibility to reuse data and samples extremely difficult. Interesting proposals coming from the Anglo-Saxon world regard the possibility for participants to establish, through Information and Communication Technologies (ICT), an ongoing, bidirectional communication with biobank institutions to refresh or withdraw their consent for new research projects (*Stein & Terry, 2013*; *Kaye et al., 2015*). In such a dynamic form of consent, the authorization of individuals to handle their personal data could travel with the same datasets containing biological and personal data (*Terry et al., 2013*). The dynamic consent approach could be conceived as a way to preserve the individual right to decide autonomously after having received detailed explanations about the biobank sharing policies (participants could be provided with as much information as they want concerning the aims of the projects aims and the methods used) and, at the same time, as a way to protect individuals' privacy (each participant is free to handle and authorize flows of personal data and to know regulations on data protection). Nevertheless, the dynamic consent approach and strict laws on data protection are not useful in all cases. Indeed, privacy laws and the rising attention towards individual rights can hinder the broad informed consent model and, overall, can hinder those bio-repositories that have been established to protect collective rights such as public health. For instance, cancer registries or retrospective studies could be damaged by the strict rules on privacy proposed by the European Parliament resolution (12 March 2014) which refers to the need to ask participants for their consent for every new research project involving their data and samples (*Casali, 2014*). At European level, the ongoing discussion on consent needed for medical research has involved most of the biobanks in which broad consent has been used to involve participants. The proposed Data Protection Regulation, if transposed into law, will constitute a challenge for biobanks and the scientific community will have to adapt to the new European regulations (*Hallinan & Friedewald, 2015*; *Lucivero et al., 2015*). On the one hand, the scientific community might use, when possible, ICTs to ask for an individual, specific consent for every new project involving personal data. On the other hand, lawyers, policy makers and experts might release the need of a specific consent, for research projects where open, blanket or presumed consent has been used appropriately. This reasoning leads back to the aforementioned problem of the lack of common and standardized operating procedures (SOPs) and heterogeneity in access rules. In fact, this fragmentation not only limits the benefits of sharing for the academia, but also contributes to increasing uncertainty of prospective donors in deciding whether or not to give their contribution to the activities of biobanks. Undoubtedly, progress in biomedical

research is strictly linked to the involvement of the public in biobank activities. In short, no donors, no biobanks. But the willingness of citizens to donate biological samples and actively participate in biobank activities are in turn strictly linked to clarity when explain the importance of participating in medical research (i.e., benefits deriving from biobank research) and the manner in which biological samples and data will be used and made available to the scientific community. In short, to foster mutually productive relationships among all the stakeholders regarding biobanks, it is necessary to develop trust "understood as something which demands knowledge and consent" (*Richter, 2012*), and to produce policies that make biobanks trustworthy and sustainable institutions (*Simeon-Dubach & Watson, 2014*).

## CONCLUSIONS

In this paper, we have attempted to analyze the degree of accessibility and reusability of data and biological samples stored in research biobanks following an empirical approach. Mainly, this study suggests that, in spite of general consensus of the scientific community concerning the importance of open access of scientific resources, there are still sample and data sharing barriers among biobanks and researchers. This does not mean that all these barriers should be necessarily overcome thus leading to unrestricted access to biobank resources. In fact, some of these barriers guarantee some fundamental rights of donors (e.g., privacy, misuse prevention) so should be considered as "necessary." Therefore, bearing in mind the need to respect the donors' rights when trying to overcome the sharing barriers, the accessibility of biological resources should not be "unified" but rather go through standardized operating procedures.

Undoubtedly, this preliminary investigation needs to be continued and improved in order to support (or even to question) the results obtained. Particularly, increasing the number of surveyed biobanks and the related differences of socio-cultural contexts could help in producing a more detailed picture of sharing behaviors and their differences related to the countries where biobanks operate. Furthermore, more information could be obtained following a two-step research protocol based on quantitative approaches such as those used in the present study, and a second, more deeply focused, qualitative investigation (e.g., semi-structured interviews, focus groups and interviews) into the main issues that emerge from the first step. According to *Mertz et al. (2014)*, empirical approaches provide an opportunity to overcome the classical descriptive aim of social science methods applied in studying the scientific environment. Considering this point of view, the so-called "empirical ethics" (see *Hope, 1999*; *Molewijk et al., 2004*) may contribute to increasing the knowledge on how and in what way all the agents involved in the life cycle of biomedical research share their work.

## ACKNOWLEDGEMENTS

We wish to thank the biobanks and their anonymous members who contributed to the survey. We would also like to thank the reviewers for their helpful comments and suggestions that greatly contributed to improving the final version of the manuscript.

### Funding

The authors received no funding for this work.

### Competing Interests

The authors declare there are no competing interests.

### Author Contributions

- Marco Capocasa and Paolo Anagnostou conceived and designed the experiments, analyzed the data, wrote the paper, prepared figures and/or tables, reviewed drafts of the paper, collected the data.
- Flavio D'Abramo analyzed the data, wrote the paper, prepared figures and/or tables, reviewed drafts of the paper.
- Giulia Matteucci conceived and designed the experiments, reviewed drafts of the paper, collected the data.
- Valentina Dominici analyzed the data, reviewed drafts of the paper.
- Giovanni Destro Bisol reviewed drafts of the paper, provided critical and theoretical inputs.
- Fabrizio Rufo conceived and designed the experiments, wrote the paper, reviewed drafts of the paper, collected the data.

### Data Availability

Zenodo (DOI: 10.5281/zenodo.17098).

### Supplemental Information

Supplemental information for this article can be found online at http://dx.doi.org/10.7717/peerj.1613#supplemental-information.

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
