# Peer review of "Samples and data accessibility in research biobanks: an explorative survey"

_PeerJ, doi:10.7717/peerj.1613_

## Round 0.1 · original submission · Major Revisions

Please pay special attention to the comments of Reviewer #1.

·

Basic reporting

Acceptable

Experimental design

Not acceptable, at least at this form. It is unclear how these biobanks were selected, especially as there is no biobank that I am the head of listed there, there is not even Croatia as a country listed there. This makes me wonder if the sampling and survey sending was representative or not at all? This is puzzling, especially as you opted out for a quantitative study and then miss the critical portion of it. Why did you not use any of the existing catalogues, such as P3G or BBMRI? This would have provided much better outreach. Then, how did you select biobanks, based on which criteria? There are about 400-500 biobanks in Europe, estimated, so you need to explain how you selected those that you approached. It is all about validity of your results, if you go in quantitative study direction.

Validity of the findings

Problematic, due to the sampling design issues. It is unclear of the results are valid due to severe selection bias risk. Secondly, some biobanks are dealing with highly sensitive data, and their policies must prohibit direct access due to the high risk of identification of the subjects. Therefore it seems rather unlikely that a unified access is what we should seek out. This needs to be discussed in the paper. You are also lacking a section on the current and ongoing EU discussions related to the sample consent being sought out for every type of future analyses, what is a possible deal-breaker for biobanks in Europe. This is a very serious discussion, and you should provide more input into this in terms of biobanks context.

Additional comments

I agree with the need to resolve the current status, but you did not even provide a framework for the most basic input, which is creation of a general repository of biobanks across the world (such as BBMRI is trying to do). Your conclusions are useful at this moment, but might loose their validity very soon, as the field of biobanking is very volatile.

·

Basic reporting

• Language editing is highly recommended. Some obvious examples are marked in PDF attached, but the rest of the text needs a touch by a native speaker too.

• There are risks (as well as the benefits) in data sharing. It is not so easy to share research data from clinical trials on humans, since there are legal, privacy etc. issues. The risks should be addressed too.

Experimental design

Pg 7 Ln 16-17 „The list of biobanks has been acquired through a research using Google, Google Scholar and Pubmed as search engines. The research has been based on the following keywords: biobank”, “research biobank” and “human biobank”.“ Could you provide some more details on number of results found and criteria how you picked the results?

Validity of the findings

No Comments.

Additional comments

• Pg 2 Ln 4-5 „In fact, making bio-resources promptly accessible to all, can favour collaboration among research groups as well as multidisciplinarity.“
Too strong. Could instead of can? There is no evidence provided for such a statement. Besides, collaboration between the research groups should not be the main outcome. What about patients? Consider deleting this sentence, and addressing other, more important benefits of data sharing.
• Pg 2 Ln 8. „46 out of the 238...“ Please reformulate so the sentence does not begin with a number.
• Pg 2 Ln 7-8. „In this study we conducted a questionnaire based survey in order to investigate sample and data accessibility in research biobanks operating all over the world.“
Pg 2 Ln 11-12. „The analysis of the biobanks guidelines regarding the accessibility of their resources reveal the importance of three aspects...“ These two seem to be the aims of this study, or not? Put them together if they are.

• Pg 5 Ln 16-17 „Making bio-resources promptly accessible to all, undoubtedly provide more opportunities for collaboration and encourage multidisciplinarity.“ Once again, yes, that is one benefit, but they are not primary in medicine, patients' benefits come first. Please declare patients' interests in sharing.
• Pg 7 Ln 9 „Methods“. Replace with „Materials & Methods“. These are standard sections (https://peerj.com/about/author-instructions/).
• Pg 9 Ln 4-5 „A total of 46 biobanks operating in four continents replied to our survey invitation: 26 in Europe, 16 in America, 2 in Asia and 2 in Oceania (Table 1).“ Suggestion: it would be nice to see response rate in this sentence e.g. Out of ... biobanks, 46 (...%) responded to our survey...
• Pg 9 Ln 24 „...and operate in the United States (3) ,“ There is an extra space between citation and comma.
• Pg 10 Ln 8. Since this is the first time you use term Institutional Review Board, put the acronym (IRB) behind it.
• Pg 10 Ln 10-12 „Open consent seems to be the most utilized manner to involve donors, whereas informed consent results to be less adopted.“ Not clear, please reformulate.
• Pg 10 Ln 12-13 „Some biobanks provide information on privacy issues describing, in most cases, the anonymized character of the personal data.“ Did you mean: type of anonymization? Not clear, please reformulate.
• Pg 14 Ln 3 Please start the first paragraph of Discussion with the most important findings of your study. Second paragraph should introduce the limitations of the study.
• Pg 14 Ln 18-20 „Starting from these premises, we conducted a questionnaire-based survey in order to shed light on how and at what level data and biological samples stored in research biobanks are accessible and reusable.“ This sentence belongs to Materials & Methods, not Discussion.
• Pg 15 Ln 19-20 „Secondly, the role research funds, public or private, in the establishment of fruitful collaborations between biobanks and research labs.“ The main verb is missing, please, reformulate.
• Pg 15 Ln 20-21 „Public research seems to be preferred over studies granted...“ Suggestion: Publicly funded research.
• Pg 17 Ln 13-18 „Anagnostou et al (2015) analyzing the data sharing rates in human paleogenetics showed that among researchers of this research field, data sharing is indeed a common practice. In fact, almost the totality of data were actually immediately available (97.6% of datasets). According to the authors it seems that this good sharing practice is to be attributed much more to a general awareness of the importance of openness and transparency for scientific progress rather than comply with norms or expectations of any scientific reward.“ Unnecessary and inadequate citation. Certainly no legal or privacy issues in sharing the data of people deceased long ago! Suggestion: delete. Please acknowledge the risks in sharing living humans' data. Reading suggestion: http://books.nap.edu/openbook.php?record_id=18998&page=33
• Pg 18 Ln 4-7„In this respect, standards to be applied to biobanks often fall under national and communitarian laws regulating trade, where often real and proper commercial battles are present and where bilateral agreements taken on global scale could influence or divert local interests and economies (e.g. the application of the Transatlantic Trade and Investment Partnership within national health services).“ The point not clear, please reformulate.
• Pg 18 Ln 11-12„Comparing the information obtained from the biobanks, neither strategies nor standards result to be shared among these institutions.“ Not clear, please reformulate.
• Pg 19 Ln2 „As stated by Jane Kaye et al“ full stop missing.
• Pg 19 Ln6 „difficult .“ There is an extra space in front of full stop.
• Pg 20 Ln 12 „Concluding remarks“. Replace with „Conclusions“.
• Pg 21 Ln 1 „According to Mertz et al (2014)“ full stop missing.
• Pg 21 Ln 11-12 „Acknowledgements We would like to thank the biobanks and their anonymous members who contributed to the survey. This work was supported by the Istituto Italiano di Antropologia (http://www.isita-org.com/).“ Acknowledgements: Should not be used to acknowledge funders – that information will appear in a separate Funding Statement on the published paper (https://peerj.com/about/author-instructions/).
• Pg 31 1st row „Sud Africa“. Replace with „South“?

·

Basic reporting

Generally this is a very interesting paper, pointing to a very important issue of biobanks specimen and data sharing and points to some obstacles.

The manuscript more or less follows the required structure. You may wish to add headings to the abstract. Headings might be installed as follows: Background -line 2-6; Methods line 7. the last part might require some re-org as Results are in lines 8-15 and 17-18 while Discussion/conclusions lines 15-16 and 18-20. You commented major results irght away, which might make it easier to read, but if needed, you can either give a common heading or you might re-organize this part to put all results together nad all discussion/conclusion together.
By the way I find it very good that results take the a large part of the abstract .

English language editing is needed, both grammar and sometimes to ensure clarity. I indicated in some parts of the manuscript, but there is more.

More edit suggestions are below.

Experimental design

The reserach question and the methods are well described providing valuable details. However, it would be useful to know whether you piloted/tested the questionnaire.
Also, the launch and number of reminders is not clear. Pls see: P 8, line 6 and 7. If I read well you launched your survey on 18th April and sent 3 reminders: 28th April, and 5th and 19th May. Is that so, or did you launch it earlier and sent 4 reminders? By the way you would have probably.

Detailed comments are given below following the text..

Validity of the findings

Few edits suggested below following the manuscript page by page and line by line.

Additional comments

Page 2, Line 11 & 12, (Abstract). You indicate that you analysed biobanks’ guidelines. Please clarify whether you analysed guidelines of the whole sample ie 238 biobanks or just guidelines of the 46 biobanks that responded to your survey.

P4, line 10: clarify that you or Macilotti means by “data carriers” so that reader can understand ie why bio samples in Europe are considered as data carriers.

P6, line 12-15. You state that the Belmont report is usually followed. Usually where? US? US and Italy, USA and ? Also, what about DOH (Declaration of Helsinki, as international or some national legal frameworks that you mention later on-see p13, line 10-13.

P8, line 18: you classified answers in languages other than English as elusive. Do you mean E or English or Italian? See your page 11, line 18.

This is just a comment : P8.line 19 &20: you indicate you deposited data in Zenodo. It is great that you deposited data in Zenodo, but actually you deposited this manuscript, which I find sort of strange.

P9. 8-10 : ….”more than half of sampled” - half of 238 or 46?, please be precise. Actually you can define it once for the whole manuscript. Namely you could probably figure out the funding of biobanks by analyzing their websites.

P9.line 9: “whereas … make use of private funds…: “please specify: are they private for profit and/or non profit? If both profit and non profit are in this group, please present the % separately for each group.
P9 line 14: the low number of institutions surveyed. Please edit. You surveyed 238 institutions (ie biobanks) but only 46 responded to your survey.
P13, line13-15: please precise from which country or region these legal frameworks come.
P13, line 10-13 it is not clear from your table and report, whether the same legal framework is used for bio specimen and for data by the same biobanks, the table shows these info separately . Please clarify

P13 line 17&18: “only two biobanks provide the specific weblink. Please clarify whether you got this link via survey you by browsing biobanks' websites?
P17, line 22-24: you wrote: “characteristics of public research are more and more similar to those of private commercial science:” Are they? Is this your take based on this survey or you can provide the literature?

P19, line 1 & 2: “the original formulations of the Declaration of Helsinki” . Reformulate/ clarify: DOH is revised every 3 years and the latest versions replaces previous. So when Kaye wrote her paper in 2015 she probably referred to the 2013 version. The "original formulation" might lead the reader to think about the 1964 version which is out of use.

---

## Round 0.2 · Minor Revisions

The two reviewers positively commented on the revised article, and only minor changes are needed. However, both reviewers and myself noticed a serious problem of language and style. Please pay special attention to this in the revised manuscript.

·

Basic reporting

Introduction is too long, needs condensing and more clear line of argumentation.

Experimental design

Pg 7, Ln 18-22 “Furtermore, we performed a further refinement adding the term “human” and obtained 4,500 results with Google, 284 with Google Scholar and 19 with Pubmed. The selection of the biobanks included in our dataset was based on the results of this latter search. This keyword based procedure was adopted in order to select a random sample of biobanks that could be easily found by anyone (researchers and public).”

It is still not clear how 238 biobanks were selected from “4,500 (Google), 284 (Google Scholar) and 19 (Pubmed)”. Given the sampling bias commented by Reviewer 1, this issue has to be clarified precisely.

Validity of the findings

No Comments.

Additional comments

It would be useful to perform a spell-check.

·

Basic reporting

The manuscript is really improved. Authors thoroughly dealt with all comments. Important topic.
I have only one comment beside language/copy editing. Line 413: It is not clear what do you mean by characteristics? Does it mean they are both corrupted or? Namely in your rebuttal you cited Krimsky. Also this sentence would merit language editing.

I suggest overall language editing/ or copy editing. Here are few examples of errors:
line 128: erroneously edited from correct "publicly" to incorrect "public" funding
line 151: it should read neither.... nor (you wrote nor....neither)
lines 321-324 the word agreement is somehow lost at the end of the sentence
322-326: mixed past tense and present tense in the same sentence: "highlighted..... provided..." Also no need for " in the answer" .. I suggest the past tense for both. Perhaps his part of he sentence could read: .."only two biobanks (7,7%) provided the web link to the legal framework documents"

Experimental design

ok

Validity of the findings

ok

Additional comments

Congratulate, well addressed comments, manuscript improved , reads much better.. I have only one comment on the substance ,please see in the Basic reporting. Beside that I advise you now edit, to correct language errors.

---

## Round 0.3 · accepted · Accept

Thank you for the latest revision. The manuscript is now significantly improved.